# The Impact of Digital Development on Port Security Resilience—An Empirical Study from Chinese Provinces

**Xiaohong Ren [1,*], Jia Shen [1], Zhitao Feng [2], Xihuai Wang [3] and Kaige An [4]**

1   School of Economics and Management, Chongqing Jiaotong University, Chongqing 400074, China; shenj187@163.com
2   School of Transportation, Chongqing Jiaotong University, Chongqing 400074, China
3   School of Architecture and Urban Planning, Chongqing Jiaotong University, Chongqing 400074, China
4   School of Civil Engineering, Chongqing University of Arts and Sciences, Chongqing 402160, China
*   Correspondence: renxh814@126.com

**Abstract:** As the world transitions into the post-pandemic era, ports in various countries are experiencing increased activity, leading to significant challenges in ensuring traffic safety within port jurisdictions. It is essential to explore methods to improve port security resilience to maintain regular economic and trade exchanges. This article examines the influence of digital development on port security resilience. Firstly, the provincial digital development level score is objectively evaluated from the perspective of digital infrastructure, digital application, and digital industry development. Secondly, the port safety resilience score is assessed considering water traffic safety, rescue and recovery capabilities, and operational efficiency. Lastly, the focus is on 16 provinces in China's coastal and Yangtze River Economic Belt, establishing provincial panel data from 2010 to 2019, and empirically analyzing the direct impact of digital development on the resilience of port security. The results indicate that digital development enhances the resilience of port security, with significant heterogeneity and long-term effects observed.

**Keywords:** digital development; port security resilience; entropy weight method; long-term effect

## 1. Introduction

The COVID-19 pandemic has highlighted the need for global economic resilience [1]. Ports, essential connectors in global trade, must adapt quickly to recover from risk shocks [2]. However, ports face various risks [3] like natural disasters, cyberattacks, strikes, hazardous goods transport, and water traffic safety, all impacting port resilience. Traffic accidents in ports can greatly reduce operational efficiency, disrupt waterway functioning, damage infrastructure, and undermine the port's transportation role [4]. For instance, incidents like ship collisions in Zhangzhou Port, Valencia Port, and Haiphong Port have caused significant damage. Utilizing digital tools to report ship information and improve navigation plans through Vessel Traffic Management Centers (VTS) can help prevent such accidents [5]. Integrating communication technologies like microwave, 5G, and Wi-Fi 6 to connect people, vehicles, ships, objects, and goods within the port can enhance overall intelligence, ultimately minimizing accidents and improving safety performance.

Previous studies have delved into the concept of port resilience, emphasizing ports' capacity to withstand and recover from various risks [6]. While research has explored port resilience in the context of epidemics, natural disasters, man-made incidents, and cyberattacks, there is a notable gap in understanding water traffic safety. Despite the multitude of risks that ports face, traffic safety accidents are both common and probable [7]. Developing a comprehensive port security resilience index could enhance the breadth of research on port resilience and enable a more in-depth analysis of factors influencing resilience, such as the influence of digital development levels on port security resilience. Investigating the impact of digitalization on port security resilience not only helps ports address various

risk challenges, enhance safety management levels, and improve operational efficiency, but also supports the sustainable development of ports. This research contributes significantly to strengthening the stability and resilience of the global supply chain. The topic holds practical and theoretical significance for understanding and utilizing digital technologies to safeguard and enhance port security resilience.

## 2. Literature Review

This article aims to explore the influence of digital advancements on the resilience of port safety. The study will analyze pertinent literature from various angles, such as resilience and port security, digital development, and research on digitalization within the port industry. Additionally, a critical assessment of these studies and an overview of research gaps will be presented.

### 2.1. Resilience and Port Security

Originally introduced in the field of ecology by Holling [8], resilience refers to the persistence of relationships within systems and acts as a metric for these systems' capacity to withstand external shocks without undergoing significant changes. Over time, the concept of resilience has been extended to various disciplines including sociology [9,10], psychology [11,12], economics [13,14], and transportation [15–17]. In these studies, resilience is commonly defined as a system's ability to adapt to and recover from risk shocks. Despite its widespread application, resilience remains a complex and abstract concept, prompting discussions on specific quantitative methodologies. Previous research often measures resilience in the context of specific events like earthquakes, hurricanes, fires, and floods [18], establishing one or more measurement frameworks. Various methods such as entropy theory, the system function curve, and the Bayesian network model are employed to quantify resilience [19]. Recent studies propose the use of stochastic mixed-integer programming to assess the resilience of entire systems, distinguishing between resilience and reliability. This approach suggests that system resilience, akin to a system metaphor, should be evaluated through a mathematical aggregation of multiple indicators [20].

In the field of port research, it has been noted that long-term experience shows a growing diversity of risks facing port development. Therefore, it is crucial to investigate the factors that threaten port security [21]. The recent emergence of the new coronavirus pandemic (COVID-19) has resulted in the infection of numerous port personnel through viral transmission, posing a threat to port operation safety and spreading globally through supply chain transmission [22]. In light of this, some scholars have developed a resilience assessment framework using the Bayesian network model and applied it to conduct specific port resilience assessments [23]. Others have explored the resilience and clustering characteristics of ports as nodes within shipping networks [24]. A recent article has taken a macro perspective by using the entropy weight method to establish a port resilience index system that considers the interests of multiple stakeholders, and has further examined the relationship between resilience and port performance [25]. Previous discussions on port resilience have focused on specific scenarios [26,27], but ultimately, risks such as ship collisions, reefs, and strandings in port areas are more common occurrences. It is important to note that resilience and security are closely intertwined, and measuring resilience can greatly enhance port security, ensuring the smooth functioning of ports. The current research has not thoroughly examined the inherent relationship between port safety and resilience, especially in the development of response strategies and the improvement of port emergency response capabilities. Therefore, it is crucial to conduct a detailed analysis of the various risks that ports face to establish a comprehensive set of indicators for port safety resilience. This system should not only effectively evaluate the safety of ports but also demonstrate their ability to adapt and recover in the face of different external challenges.

### 2.2. Digital Development

The continuous evolution and integration of technologies such as the Internet, 5G, big data, and cloud computing have had a profound impact on the social economy. Research

suggests that governments can use digital means to promote high-quality economic development [28], facilitate the transformation and upgrading of the manufacturing industry [29], enhance environmental governance performance [30], and advance rural governance transformation [31]. In the realm of transportation, studies have indicated that digitization can reduce information asymmetry, build mutual trust [32], lower logistics costs, and support the implementation of green logistics [33]. Additionally, digital twin technology and artificial intelligence have shown significant advantages in classifying transportation infrastructure and managing traffic spatial information networks [34]. Digital development also plays a role in reducing carbon emissions in the transport sector. Jiao and Zhang [35] found that digitization leads to emission reduction benefits, and green technology innovation can substantially decrease transportation carbon emissions in the long term. Pu and Lam [36] noted that the greenhouse gas emissions from digital documents during transportation are lower than those from paper documents. Moreover, digital technology can greatly enhance traffic safety by using technologies like artificial intelligence, the Internet of Things, and image processing to automatically collect and analyze data, effectively reducing road accident rates [37].

### 2.3. Research on Digitalization within the Port Industry

The impact of digital development on socioeconomic aspects is significant, particularly in the realm of ports. Ports play a critical role in the supply chain, and as political, economic, social, and environmental factors gravitate towards digitalization and sustainable growth, there is a necessity for ports to transition into smart ports aligning with Industry 4.0 standards [38]. By deeply integrating digital technologies like the Internet of Things, big data, cloud computing, and artificial intelligence, smart ports can enhance their operational capabilities intelligently and optimize the allocation of resources for port operations [39]. This transformation will increase the competitiveness of ports in tackling future challenges [40]. Moreover, fostering the digital evolution of ports necessitates collaborative efforts from all stakeholders, particularly port management, enterprise operators, and suppliers, to overcome communication barriers and bolster interaction [41].

The digital transformation of port and shipping enterprises may present challenges initially, but it ultimately enhances their operational resilience in the long term [42]. Major shipping companies have embraced digitalization to enhance cost efficiency and competitiveness in meeting customer demands [43]. Traditional shipping logistics firms may face resistance in adopting technologies like blockchain, cloud data, the Internet of Things, and big data analytics [44,45]. However, digital technologies, including software platforms, hardware solutions, and data analytics, will facilitate the greater integration of port enterprises [46]. Notably, the advancement of blockchain technology plays a critical role in ensuring the stability of the maritime supply chain [47] and driving transformation across the industry [48].

### 2.4. Review of the Literature and Research Gap

The existing research has primarily focused on defining and exploring port resilience from individual perspectives, aiming to identify key factors that can improve resilience and discussing ways to maintain normal port operations by enhancing safety performance. While there is a strong connection between port safety and resilience, few scholars have integrated these concepts in the current research. A comprehensive framework for establishing universally applicable port safety resilience indicators is lacking, making it challenging to assess and enhance port resilience effectively. Furthermore, as society moves towards digitization, digital development plays a crucial role in driving the transformation and upgrading of ports. Despite the opportunities digital progression offers to port operations, there is limited research on how it specifically impacts port safety resilience mechanisms. Further exploration in this area is needed to understand how digital development enhances port safety resilience and how digital tools can better manage and mitigate risks and challenges faced by ports. This study examines traffic safety within port jurisdictions, focusing on establishing a port safety resilience (RES) indicator system using the entropy weight

method. The system considers water traffic safety, rescue and recovery capability, and operational capability. Additionally, a digital development level (DIG) evaluation system is created. Using provincial panel data from China spanning 2010 to 2019, annual scores are calculated for port safety resilience and digital development levels. The study explores the direct impact of DIG on RES, analyzing the effects of DIG on RES in terms of regional heterogeneity and long-term outcomes.

The article is structured as follows: Section 2 reviews the relevant literature and identifies research gaps, Section 3 outlines data sources and research methodology, Section 4 presents empirical analysis, endogeneity considerations, robustness tests, and extension analysis, and Section 5 discusses research findings, policy recommendations, and future research directions.

## 3. Materials and Methods

This section introduces the research area, establishes the indicator system, describes variables, formulates models, and identifies data sources. The research flow chart is shown in Figure 1.

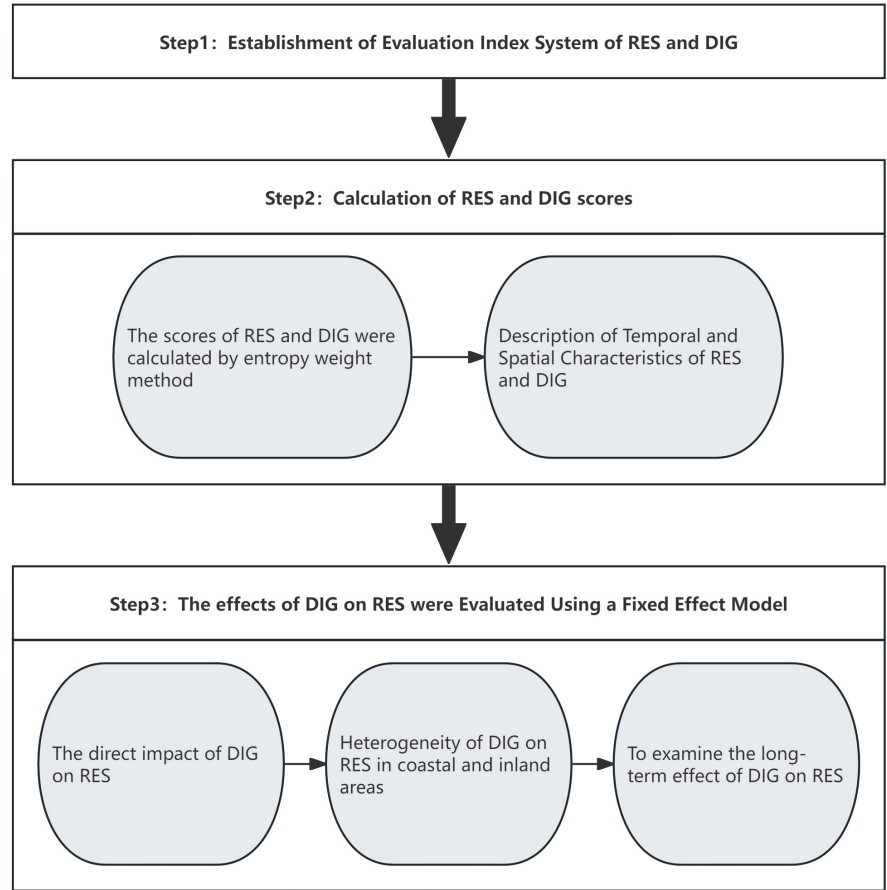

**Figure 1.** Flow chart of the research.

### 3.1. Research Area

The study covers a total of 16 provinces along the Chinese coast (Hebei, Tianjin, Shandong, Jiangsu, Shanghai, Zhejiang, Fujian, Guangdong, Guangxi, and Hainan) and the economic belt of the Yangtze River (Anhui, Jiangxi, Hubei, Hunan, Chongqing, and Sichuan) for analysis (see Figure 2). These areas represent a significant portion of China's coastline and inland waterway transportation, providing a comprehensive view of the country's situation. Hong Kong, Macau, and Taiwan were not considered in the study due to data availability constraints. Additionally, Liaoning, Guizhou, and Yunnan were excluded from the analysis because of insufficient data.

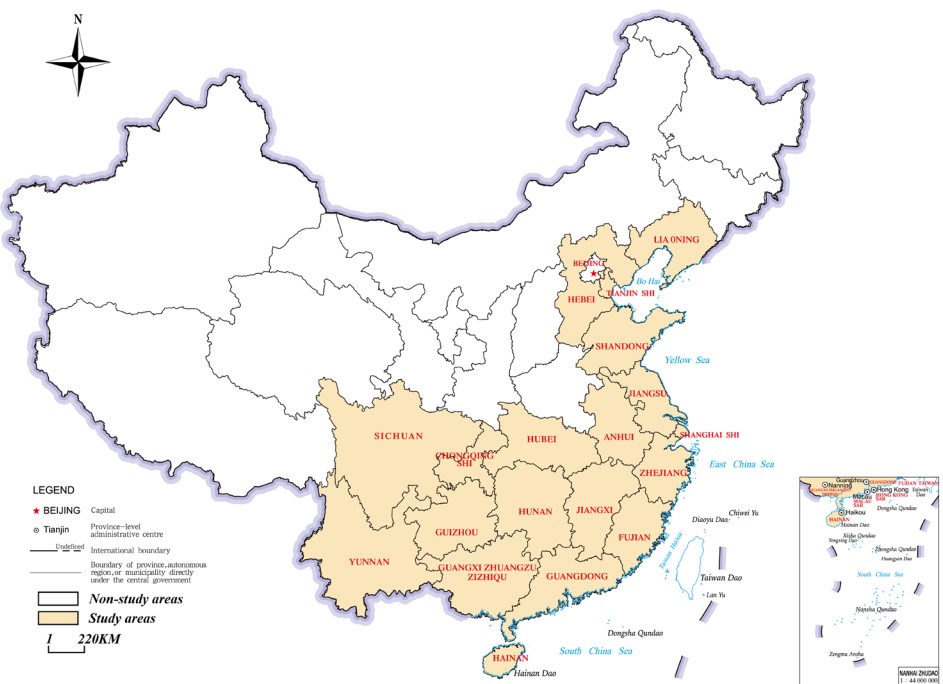

**Figure 2.** Spatial distribution of the research areas.

*3.2. Index System*

Based on the definition of resilience and considering the relevance and comparability of data, the resilience of port safety is categorized into three components: water traffic safety, rescue and recovery capacity, and operational capacity. This pertains to the capability to promptly evacuate and rescue individuals and maintain basic operations in the event of water traffic safety incidents within port jurisdictions. The RES evaluation index system is detailed in Table 1. Presently, there is no standardized method within the academic community to measure the extent of urban digitalization. While some studies rely on a single indicator to gauge digital development, this approach may not offer a comprehensive view, given that digital development is a multifaceted process that influences economic growth and societal progress. Consequently, certain researchers have started to assess digital development from various perspectives. Building upon the research conducted by Pang et al. in 2021 [49], this paper evaluates the level of digital development across three dimensions: digital infrastructure, digital application, and digital industry development. The DIG evaluation index system is outlined in Table 2.

**Table 1.** Port security resilience index system.

| Primary Index | Secondary Index | Tertiary Index | Unit | Index Type | Weight |
|---|---|---|---|---|---|
| Port security resilience level | Water traffic safety | Number of water traffic accidents | Time | - | 0.014 |
| | | Number of dead/missing persons | Number of people | - | 0.010 |
| | | Number of people employed in the water transport industry | Number of people | + | 0.050 |
| | Rescue recovery capability | Investment in fixed assets in transportation, warehousing and postal services | Hundred million yuan | + | 0.069 |
| | | Highway density | km/km² | + | 0.045 |
| | | Railway density | km/km² | + | 0.082 |
| | | Number of beds in medical institutions | Ten thousand pieces | + | 0.059 |
| | Operational capability | Cargo throughput | Ten thousand tons | + | 0.069 |
| | | Foreign trade throughput | Ten thousand tons | + | 0.175 |
| | | Container throughput | Ten thousand tons | + | 0.169 |
| | | Length of wharf berth | M | + | 0.117 |
| | | Number of dock berths | Number | + | 0.142 |

"-" indicates a negative indicator, "+" indicates a positive indicator.

**Table 2.** Digital development level index system.

| Primary Index | Secondary Index | Tertiary Index | Unit | Index Type | Weight |
|---|---|---|---|---|---|
| Digital development level | Digital infrastructure | Toll cable density | Ten thousand kilometers | + | 0.073 |
| | | Per capita mobile phone exchange capacity | - | + | 0.084 |
| | | Number of broadband Internet access ports per capita | Number | + | 0.078 |
| | | Internet penetration | % | + | 0.066 |
| | | Mobile phone penetration | Number of people | + | 0.043 |
| | Digital application | Number of websites owned by each enterprise | Number/hundred | + | 0.023 |
| | | Per capita express business volume | Number of packages | + | 0.319 |
| | | Digital industry employment accounted for the proportion of urban units | % | + | 0.191 |
| | Digital industry development | The proportion of fixed asset investment in digital industry in the whole society | % | + | 0.019 |
| | | Share of digital industry revenue in gross regional product | % | + | 0.105 |

"-" indicates a negative indicator, "+" indicates a positive indicator.

### 3.3. Variable Description

### 3.3.1. Explained Variables

Provincial port safety resilience level (RES): Based on the index system in Table 1, the entropy method is used to obtain the provincial port safety resilience level. The steps are as follows:

First, the collected indicator data are normalized and standardized. When normalized, the data will have a value of 0; therefore, we use 0.0001 in place of data with a value of 0. The processing method for positive indicators is shown in Equation (1) and the processing method for negative indicators is shown in Equation (2).

$$Z_{ij} = \frac{X_{ij} - \min X_{ij}}{\max X_{ij} - \min X_{ij}}, i = 1, 2, \ldots, n; j = 1, 2, \ldots, m \tag{1}$$

$$Z_{ij} = \frac{\max X_{ij} - X_{ij}}{\max X_{ij} - \min X_{ij}}, i = 1, 2, \ldots, n; j = 1, 2, \ldots, m \tag{2}$$

Secondly, the information entropy and weight of the standardized data are calculated. For an indicator $j$, the greater the difference in the $Z_{ij}$ value, the greater the effect of this indicator on the evaluated object. That is, the more information it provides to the evaluated object, the greater the corresponding weight, and vice versa. The calculation process is shown in Equations (3)–(5).

Calculate information entropy:

$$p_{ij} = \frac{Z_{ij}}{\sum\limits_{i=1}^{n} Z_{ij}}, i = 1, 2, \ldots, n; j = 1, 2, \ldots, m \tag{3}$$

$$e_j = -\frac{1}{\ln n} \sum p_{ij} \ln p_{ij}, i = 1, 2, \ldots, n; j = 1, 2, \ldots, m \tag{4}$$

Calculate the weight:

$$W_j = \frac{(1 - e_j)}{\sum\limits_{j=1}^{m} (1 - e_j)}, j = 1, 2, \ldots, m \tag{5}$$

In all of the above formulas, $X_{ij}$ is the $j$th original index value in $i$th year in a system, $Z_{ij}$ is the standardized value, and $\max X_{ij}$ and $\min X_{ij}$ are the maximum and minimum values of the $j$th indicator; $e_j$ is the information entropy value of the $j$th indicator; $W_j$ is the weight of the $j$th indicator.

Finally, the weighted summation formula is used to calculate the complete score or the level of evaluation.

### 3.3.2. Core Explanatory Variables

Digitization Development Level Index (DIG): Based on the index system in Table 1, the digitization development level is obtained by the entropy method, and the calculation procedure is the same as in RES.

The RES and DIG score levels obtained using the entropy weight method are shown in Tables 3 and 4.

**Table 3.** The score of RES.

| Province | 2010 | 2011 | 2012 | 2013 | 2014 | 2015 | 2016 | 2017 | 2018 | 2019 |
|---|---|---|---|---|---|---|---|---|---|---|
| HeBei | 0.278 | 0.295 | 0.319 | 0.336 | 0.344 | 0.346 | 0.357 | 0.369 | 0.367 | 0.364 |
| TianJin | 0.454 | 0.487 | 0.496 | 0.504 | 0.534 | 0.521 | 0.520 | 0.532 | 0.518 | 0.521 |
| ShanDong | 0.365 | 0.386 | 0.410 | 0.422 | 0.437 | 0.453 | 0.465 | 0.488 | 0.509 | 0.534 |
| JiangSu | 0.392 | 0.414 | 0.412 | 0.438 | 0.455 | 0.452 | 0.463 | 0.465 | 0.435 | 0.450 |
| ShangHai | 0.360 | 0.368 | 0.381 | 0.375 | 0.379 | 0.389 | 0.392 | 0.437 | 0.436 | 0.433 |
| ZheJiang | 0.297 | 0.320 | 0.328 | 0.327 | 0.340 | 0.349 | 0.347 | 0.363 | 0.374 | 0.388 |
| FuJian | 0.206 | 0.219 | 0.233 | 0.245 | 0.263 | 0.272 | 0.274 | 0.260 | 0.274 | 0.285 |
| GuangDong | 0.300 | 0.341 | 0.354 | 0.366 | 0.378 | 0.387 | 0.396 | 0.415 | 0.432 | 0.446 |
| GuangXi | 0.133 | 0.144 | 0.146 | 0.157 | 0.164 | 0.166 | 0.174 | 0.177 | 0.191 | 0.199 |
| HaiNan | 0.133 | 0.143 | 0.149 | 0.148 | 0.151 | 0.157 | 0.147 | 0.157 | 0.165 | 0.175 |
| AnHui | 0.136 | 0.143 | 0.144 | 0.155 | 0.163 | 0.171 | 0.182 | 0.186 | 0.189 | 0.199 |
| JiangXi | 0.113 | 0.119 | 0.123 | 0.131 | 0.144 | 0.149 | 0.149 | 0.147 | 0.135 | 0.132 |
| HuBei | 0.167 | 0.172 | 0.177 | 0.189 | 0.197 | 0.209 | 0.215 | 0.204 | 0.202 | 0.206 |
| HuNan | 0.140 | 0.144 | 0.145 | 0.152 | 0.166 | 0.174 | 0.182 | 0.187 | 0.177 | 0.179 |
| ChongQing | 0.173 | 0.185 | 0.174 | 0.184 | 0.139 | 0.151 | 0.156 | 0.167 | 0.175 | 0.187 |
| SiChuan | 0.123 | 0.135 | 0.155 | 0.152 | 0.169 | 0.176 | 0.187 | 0.197 | 0.199 | 0.203 |

**Table 4.** The score of DIG.

| Province | 2010 | 2011 | 2012 | 2013 | 2014 | 2015 | 2016 | 2017 | 2018 | 2019 |
|---|---|---|---|---|---|---|---|---|---|---|
| HeBei | 0.088 | 0.101 | 0.121 | 0.135 | 0.143 | 0.159 | 0.183 | 0.207 | 0.226 | 0.248 |
| TianJin | 0.124 | 0.141 | 0.160 | 0.165 | 0.183 | 0.210 | 0.257 | 0.291 | 0.328 | 0.370 |
| ShanDong | 0.104 | 0.123 | 0.137 | 0.161 | 0.181 | 0.205 | 0.229 | 0.244 | 0.265 | 0.286 |
| JiangSu | 0.163 | 0.183 | 0.210 | 0.239 | 0.261 | 0.294 | 0.320 | 0.346 | 0.398 | 0.443 |
| ShangHai | 0.259 | 0.266 | 0.321 | 0.361 | 0.411 | 0.451 | 0.524 | 0.574 | 0.602 | 0.607 |
| ZheJiang | 0.148 | 0.170 | 0.203 | 0.239 | 0.279 | 0.352 | 0.411 | 0.496 | 0.565 | 0.650 |
| FuJian | 0.155 | 0.167 | 0.191 | 0.199 | 0.223 | 0.258 | 0.283 | 0.311 | 0.345 | 0.368 |
| GuangDong | 0.213 | 0.239 | 0.266 | 0.293 | 0.322 | 0.361 | 0.415 | 0.455 | 0.511 | 0.567 |
| GuangXi | 0.052 | 0.061 | 0.071 | 0.080 | 0.085 | 0.096 | 0.112 | 0.147 | 0.177 | 0.187 |
| HaiNan | 0.067 | 0.080 | 0.088 | 0.096 | 0.103 | 0.132 | 0.152 | 0.169 | 0.203 | 0.216 |
| AnHui | 0.068 | 0.079 | 0.092 | 0.103 | 0.114 | 0.142 | 0.160 | 0.180 | 0.194 | 0.211 |
| JiangXi | 0.062 | 0.068 | 0.077 | 0.086 | 0.093 | 0.109 | 0.120 | 0.136 | 0.152 | 0.169 |
| HuBei | 0.066 | 0.083 | 0.102 | 0.122 | 0.138 | 0.161 | 0.184 | 0.213 | 0.249 | 0.290 |
| HuNan | 0.164 | 0.174 | 0.183 | 0.205 | 0.228 | 0.246 | 0.265 | 0.322 | 0.353 | 0.333 |
| ChongQing | 0.111 | 0.110 | 0.135 | 0.151 | 0.167 | 0.191 | 0.214 | 0.238 | 0.266 | 0.296 |
| SiChuan | 0.105 | 0.124 | 0.139 | 0.174 | 0.191 | 0.217 | 0.237 | 0.259 | 0.276 | 0.306 |

### 3.3.3. Control Variables

Human Resources Index (People): This index is obtained from the logarithm of the number of general college students, which can reflect the reserve of human resources of the province.

Economic Openness Index (Open): Expressed by the proportion of total imports and exports to GDP, it reflects the degree of foreign exchange and openness of the region.

Science and Technology Development Concern Index (Science): This index is expressed as the proportion of science and technology expenditure in the regional general public

budget expenditure, reflecting the importance of science and technology development for the local government.

Social Investment Index (Investment): Expressed by the proportion of fixed asset investment in the GDP of the whole society, this index can reflect the efficiency of regional investment.

*3.4. Data Source*

The above data can be obtained by referring to national and local statistical yearbooks, consulting the safety departments of the major maritime safety administrations, and the emergency management departments of provinces and municipalities directly under the central government to ensure objectivity and reliability. The statistics of each variable are shown in Table 5.

**Table 5.** Descriptive statistics of variables.

| Variables | N | Mean | SD | Min | Max |
|---|---|---|---|---|---|
| RES | 160 | 0.276 | 0.127 | 0.113 | 0.534 |
| DIG | 160 | 0.221 | 0.125 | 0.052 | 0.650 |
| People | 160 | 5.422 | 0.254 | 4.667 | 5.822 |
| Open | 160 | 0.377 | 0.332 | 0.049 | 1.457 |
| Science | 160 | 0.025 | 0.015 | 0.007 | 0.068 |
| Investment | 160 | 0.735 | 0.222 | 0.210 | 1.170 |

*3.5. Examination Statistic Models*

In order to verify the effect of DIG on RES, this paper builds the following basic model:

$$RES_{i,t} = \partial_0 + \partial_1 DIG_{i,t} + \partial_j \sum_{j=2}^{m} x_{i,t}^j + \varepsilon_{i,t} \tag{6}$$

In the above formula, $RES_{i,t}$ is explained variable of province $i$ in year $t$, and $DIG_{i,t}$ is the core explanatory variable of province $i$ in year $t$. $x_{i,t}^j (j = 1, 2, \ldots, m)$ is the $j$th control variable, and represents the remaining factors affecting $RES$. $\partial$ is the regression coefficient of the corresponding variable, and $\varepsilon_{i,t}$ is the idiosyncratic error term.

## 4. Empirical Analysis

The paragraph discusses the spatiotemporal characteristics of RES and DIG, followed by empirical analysis using baseline regression and regional heterogeneity analysis. The model's reliability is strengthened through the use of endogeneity tests and robustness checks. The long-term effects of DIG on RES are also examined.

*4.1. Temporal and Spatial Characteristics of RES and DIG*

To visually represent the evaluation system for RES and DIG, scores were plotted in Figures 3–6 to illustrate changing trends over time and space.

The research area exhibited an increasing trend in RES scores from 2010 to 2019, with the average score rising from 0.236 to 0.306. Figure 5 illustrates the spatial distribution of RES scores in 2010, 2014, and 2019, showing a positive correlation with the passage of time. Coastal provinces generally had higher scores compared to inland provinces along the Yangtze River economic belt. The average RES score for China was 0.236 in 2010, 0.276 in 2014, and 0.306 in 2019, with Tianjin and Shandong leading in RES scores, followed by Jiangsu, Shanghai, Guangdong, and Zhejiang. Sichuan and Hubei, located in the upper reaches of the Yangtze River, also ranked high among inland provinces. These findings suggest that as provinces develop resilience to port security, regional disparities become more apparent. The trend of digital innovation growth (DIG) from 2010 to 2019 is depicted in Figure 4, showing an increase from an average level of 0.122 in 2010 to 0.347 in 2019 (2.8 times higher than in 2010), indicating a rapid growth in digitization. Notably, Zhejiang's score increased by more than four times, with Jiangsu, Shanghai, and Guangdong more than doubling their scores. Figure 6 illustrates the spatial distribution of DIG scores in 2010,

2014, and 2019 in the study area, revealing that scores generally rose over the years and coastal provinces had higher scores compared to inland provinces. The average DIG score for China was 0.122 in 2010, 0.195 in 2014, and rose to 0.347 in 2019, with Shanghai and Zhejiang leading in digitalization among the regions studied, followed by Guangdong and Jiangsu. Inland provinces like Sichuan and Hunan also ranked high. Similar to the trend in port security resilience development, as digital development levels in provinces progress steadily, regional disparities are also widening.

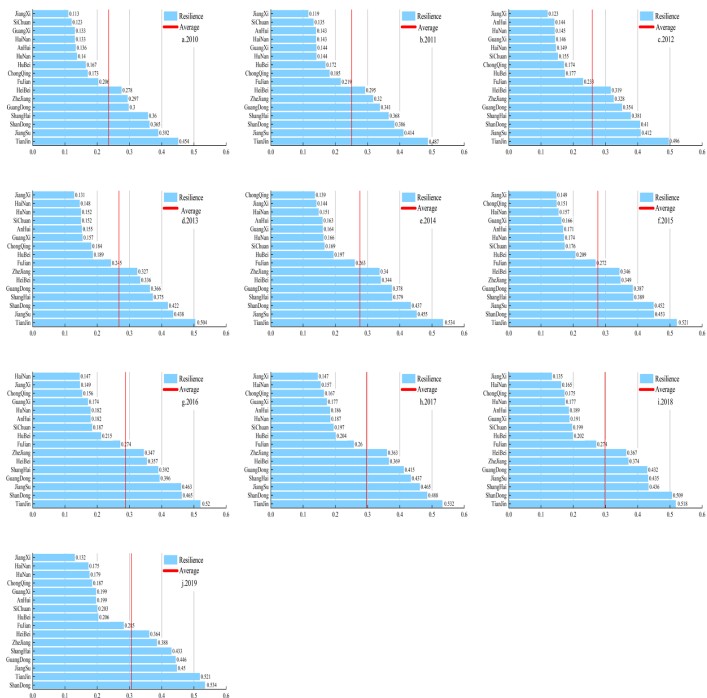

**Figure 3.** Trends of RES from 2010 to 2019.

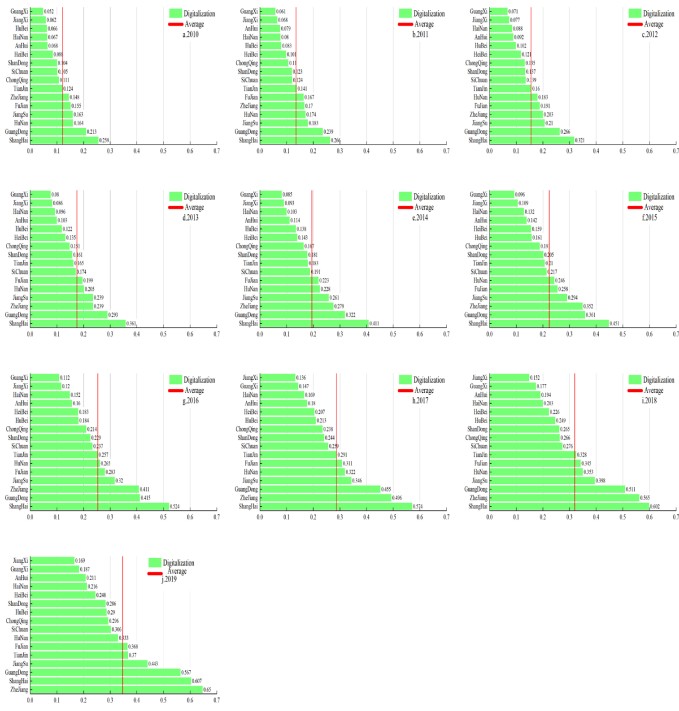

**Figure 4.** Trends of DIG from 2010 to 2019.

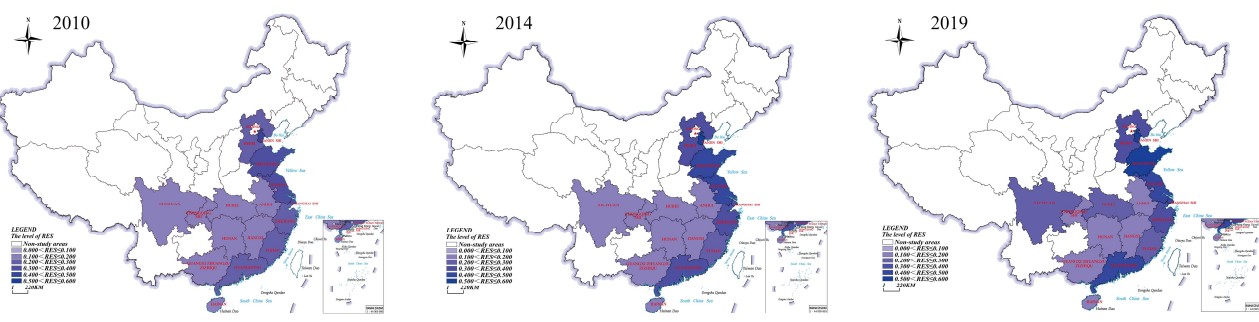

**Figure 5.** Spatial distribution of RES.

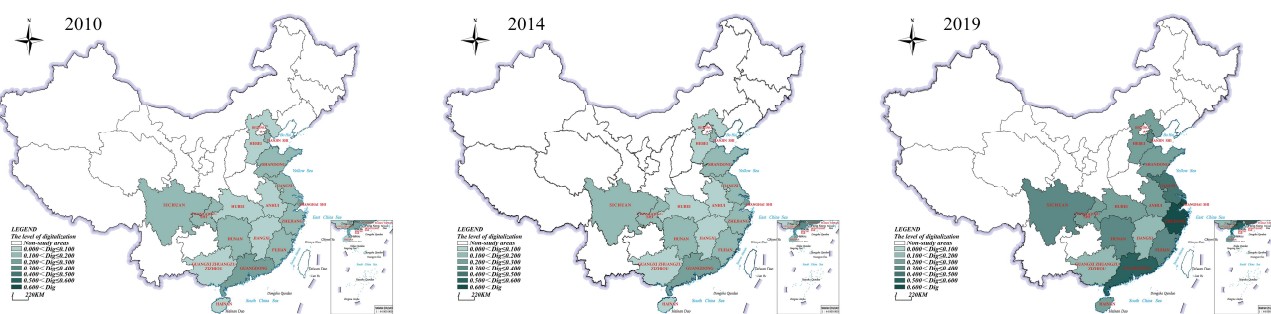

**Figure 6.** Spatial distribution of DIG.

### 4.2. Baseline Regression

Utilizing the data and methodologies outlined in the preceding sections, we established a panel fixed effect model to conduct regression analysis of Equation (6). Additionally, given that the port security resilience index and digital development level index both fall within the range of [0–1], we employed a Tobit regression model for robust estimation, with the outcomes presented in Table 6.

**Table 6.** Baseline regression result.

| Variable | (1) | (2) | (3) | (4) | (5) | (6) |
| --- | --- | --- | --- | --- | --- | --- |
| | RES | RES | RES | RES | RES | RES |
| DIG | 0.249 *** | 0.251 *** | 0.115 *** | 0.141 *** | 0.250 *** | 0.132 *** |
| | (0.040) | (0.040) | (0.035) | (0.042) | (0.017) | (0.027) |
| People | | | 0.100 * | 0.086 * | | 0.090 ** |
| | | | (0.049) | (0.050) | | (0.036) |
| Open | | | −0.071 *** | −0.048 ** | | −0.056 *** |
| | | | (0.023) | (0.024) | | (0.020) |
| Investment | | | 0.050 ** | 0.040 ** | | 0.043 *** |
| | | | (0.020) | (0.019) | | (0.016) |
| Science | | | 0.560 ** | 0.639 *** | | 0.612 *** |
| | | | (0.215) | (0.233) | | (0.226) |
| Constant | 0.221 *** | 0.221 *** | −0.314 | −0.247 | 0.221 *** | −0.268 |
| | (0.009) | (0.029) | (0.264) | (0.261) | (0.001) | (0.190) |
| Model | FE | RE | FE | RE | Tobit | Tobit |
| Sigma_u | | | | | 0.112 *** | 0.129 *** |
| | | | | | (0.017) | (0.024) |
| Sigma_e | | | | | 0.017 *** | 0.015 *** |
| | | | | | (0.001) | (0.001) |
| N | 160 | 160 | 160 | 160 | 160 | 160 |
| $R^2$ | 0.61 | 0.29 | 0.69 | 0.02 | | |

Note: Standard errors in parentheses, where *** $p < 0.01$, ** $p < 0.05$, and * $p < 0.1$.

The coefficients for DIG in Table 6 exhibit significant positive values, suggesting that advancements in digital development contribute to enhancing port security resilience. Moreover, upon accounting for the control variables, the Hausman test indicates the superiority of the fixed effect over the random effect. Consequently, this section emphasizes the fixed-effect model, specifically column (3). Based on the regression findings, a 1% increase

in DIG results in a 0.115% increase in RES. Furthermore, the proximity of the Tobit regression coefficient to the panel regression coefficient indicates the robustness of the estimation outcomes presented in this study. Regarding control variables, the coefficient of human resources (People) is 0.100 and passes the test at the significance level of 10%. The coefficient of interest in science is 0.560 and passes the test at the significance level of 5%. The regression results demonstrate that talent reserve and scientific and technological development can effectively improve the resilience of port security. In the context of digital development, all industries are actively seeking digital empowerment paths, aiming to achieve intelligent upgrading and sustainable development. In the port field, terminal production automation, intelligent berth scheduling, port logistics coordination, and security clearance facilitation can significantly improve port operation efficiency and reduce the probability of safety accidents. To realize this vision, the cultivation of talent and the development of science and technology are indispensable. The coefficient of economic openness is −0.071, with the 1% significance level test indicating that higher economic openness leads to lower RES. Higher economic openness in places like Shanghai, Guangdong, and Zhejiang often means frequent economic and trade activities, dense routes, high port operation pressure, and increased security risks. The social investment coefficient (investment) is 0.050 and passes the test at the significance level of 5%. Investment in fixed assets in the entire society reflects the structural scale and development speed of fixed asset investment, indicating the vitality of social development. This suggests that activating social assets also has a synergistic effect on improving RES.

### 4.3. Analysis of Regional Heterogeneity

As illustrated in Figures 2 and 3, distinct regional variations exist in both RES and DIG. To delve deeper into the potential heterogeneity of DIG in RES, we segmented the study areas into coastal provinces and inland provinces. A regression analysis was conducted using Equation (6), and the results are presented in Table 7.

**Table 7.** Heterogeneity analysis results.

| Variable | (Coastal) | (Coastal) | (Inland) | (Inland) |
| --- | --- | --- | --- | --- |
| | RES | RES | RES | RES |
| DIG | 0.253 *** | 0.132 *** | 0.231 *** | 0.123 * |
| | (0.021) | (0.033) | (0.030) | (0.071) |
| People | | 0.110 ** | | 0.043 |
| | | (0.051) | | (0.063) |
| Open | | −0.047 * | | −0.085 |
| | | (0.025) | | (0.060) |
| Investment | | 0.066 *** | | 0.029 |
| | | (0.022) | | (0.027) |
| Science | | 0.933 *** | | 0.478 |
| | | (0.329) | | (0.359) |
| Constant | 0.279 *** | −0.329 | 0.126 *** | −0.115 |
| | (0.006) | (0.258) | (0.005) | (−0.342) |
| Model | FE | FE | FE | FE |
| N | 100 | 100 | 60 | 60 |
| $R^2$ | 0.62 | 0.74 | 0.54 | 0.59 |

Note: Standard errors in parentheses, where *** $p < 0.01$, ** $p < 0.05$, and * $p < 0.1$.

The findings in Table 7 reveal that the influence of DIG on RES in coastal and inland provinces is notably positive, albeit with significant diversity. Particularly, DIG exerts a more pronounced impact on RES in coastal provinces compared to inland provinces. This variance can be attributed to several factors. Firstly, coastal provinces exhibit a higher level of digital development than inland provinces, with more advanced port infrastructure, facilitating a more seamless transmission of the enabling effects of digital development to the port sector. Moreover, RES development in inland provinces progresses slowly, displaying a stable trend over the past decade, indicating a relatively constant resilience over an extended period, with DIG not yielding substantial changes. Secondly, ports in coastal provinces handle a higher volume of transportation tasks, consequently increasing

the likelihood of safety incidents and resulting in more pronounced changes in RES in these regions. In contrast, inland provinces experience more tangible technical and industrial effects from digital development, leading to a clearer impact on RES. In essence, the disparity in the impact of digital development on port security resilience can be ascribed to variations in digital development levels, port construction advancement, and transportation demands between coastal and inland provinces.

### 4.4. Endogeneity and Robustness Test

To address potential endogeneity issues in the model, this study utilizes the following strategies: Firstly, regression analysis is conducted using DIG with lags of 1 and 2 periods. Secondly, the two-stage least squares (2SLS) method is applied to estimate instrumental variables with lags of 1 and 2 periods, with detailed results presented in Table 8. Additionally, to assess the robustness of the regression findings, the study employs the following approaches for robustness testing: First, outlier influence is mitigated by truncating RES and DIG at the 1% and 99% tails before conducting regression analysis. Secondly, the estimation method is altered, and panel quantile regression is utilized to address the impact of extreme values on the regression outcomes. The specific results of these robustness tests are outlined in Table 9.

**Table 8.** Endogeneity test.

| Variable | Lag 1 Period as Explanatory Variable | Lag 2 Period as Explanatory Variable | Lag 1 Period as the Instrumental Variable | Lag 2 Period as the Instrumental Variable |
|---|---|---|---|---|
| | (1) | (2) | (3) | (4) |
| L.DIG | 0.134 *** (0.028) | | | |
| L2.DIG | | 0.161 *** (0.030) | | |
| DIG | | | 0.127 *** (0.027) | 0.145 *** (0.028) |
| Constant | −0.178 (0.188) | −0.184 (0.189) | | |
| Control Variables | YES | YES | YES | YES |
| Province FE | YES | YES | YES | YES |
| N | 144 | 128 | 144 | 128 |
| R2 | 0.670 | 0.645 | 0.655 | 0.618 |

Note: Standard errors in parentheses, where *** $p < 0.01$.

**Table 9.** Robustness test.

| Variable | Winsorize | $\tau = 0.1$ | $\tau = 0.2$ | $\tau = 0.4$ | $\tau = 0.6$ | $\tau = 0.8$ | $\tau = 0.9$ |
|---|---|---|---|---|---|---|---|
| | (1) | (2) | (3) | (4) | (5) | (6) | (7) |
| DIG | 0.119 *** (0.029) | 0.083 (0.055) | 0.098 *** (0.037) | 0.112 *** (0.025) | 0.123 *** (0.023) | 0.132 *** (0.028) | 0.139 *** (0.035) |
| Constant | −0.303 (0.197) | | | | | | |
| Control Variables | YES | YES | YES | YES | YES | YES | YES |
| Province FE | YES | YES | YES | YES | YES | YES | YES |
| N | 160 | 160 | 160 | 160 | 160 | 160 | 160 |
| R$^2$ | 0.692 | | | | | | |

Note: Standard errors in parentheses, where *** $p < 0.01$.

According to the findings presented in Table 8, the level of digitization development and its lag terms (DIG, L.DIG, and L2.DIG) show a significant positive relationship with RES. This indicates that even after accounting for endogeneity issues, digitization development continues to have a positive impact on port security resilience, confirming the robustness of the conclusions drawn in this paper. Furthermore, it is noteworthy that the coefficients

for lag period 2 are larger compared to lag period 1, implying that the influence of digital development on port security resilience can endure over time. The results presented in Table 9 demonstrate that even after tail truncation treatment, the DIG coefficient remains significantly positive. Additionally, the panel quantile regression analysis reveals that the coefficient of DIG shows a gradual increase as the quantile rises from 0.1 to 0.9. This suggests that DIG has a stronger synergistic impact in high-RES sites compared to low-RES sites. It is evident from these findings that regions with high RES are typically coastal provinces with well-developed economies, solid digital infrastructure, and comprehensive port facilities. Local governments in these areas are more inclined to enhance port safety, resilience, operational efficiency, and economic stability. Consequently, the empowering effect of DIG is more pronounced in such regions.

### 4.5. Expansion Analysis

The potential long-lasting impact of digital development on the transformation of social life is explored in this study. The endogeneity test conducted in Section 4.4 hypothesizes that the influence of digital development on port security resilience may endure over time. To test this hypothesis, the researchers employed the long-term effect test method proposed by Quinn and Toyoda [50], utilizing mean data of all variables for 2, 3, and 4 years in a mean panel regression analysis. The results of this analysis are presented in Table 10.

**Table 10.** Analysis of long-term effects.

| Variable | Two-Year Average | | Three-Year Average | | Four-Year Average | |
|---|---|---|---|---|---|---|
| | (1) | (2) | (1) | (2) | (1) | (2) |
| DIG | 0.106 *** | 0.124 *** | 0.100 *** | 0.120 *** | 0.085 *** | 0.111 *** |
| | (0.028) | (0.027) | (0.029) | (0.028) | (0.032) | (0.031) |
| People | 0.097 ** | 0.090 ** | 0.096 * | 0.094 ** | 0.0840 | 0.088 * |
| | (0.045) | (0.042) | (0.049) | (0.046) | (0.0524) | (0.048) |
| Open | −0.073 *** | −0.057 *** | −0.076 *** | −0.059 *** | −0.084 *** | −0.063 *** |
| | (0.021) | (0.021) | (0.021) | (0.021) | (0.023) | (0.023) |
| Investment | 0.057 *** | 0.051 *** | 0.062 *** | 0.053 ** | 0.073 *** | 0.059 *** |
| | (0.017) | (0.017) | (0.019) | (0.018) | (0.022) | (0.021) |
| Science | 0.446 * | 0.512 ** | 0.408 * | 0.484 ** | 0.356 | 0.448 * |
| | (0.238) | (0.234) | (0.239) | (0.235) | (0.243) | (0.239) |
| Constant | −0.297 | −0.267 | −0.293 | −0.287 | −0.228 | −0.256 |
| | (0.236) | (0.222) | (0.257) | (0.239) | (0.273) | (0.252) |
| Model | FE | Tobit | FE | Tobit | FE | Tobit |
| Sigma_u | | 0.132 *** | | 0.133 *** | | 0.135 *** |
| | | (0.024) | | (0.025) | | (0.026) |
| Sigma_e | | 0.013 *** | | 0.011 *** | | 0.009 *** |
| | | (0.001) | | (0.001) | | (0.001) |
| N | 144 | 144 | 128 | 128 | 112 | 112 |
| $R^2$ | 0.715 | | 0.745 | | 0.767 | |

Note: Standard errors in parentheses, where *** $p < 0.01$, ** $p < 0.05$, and * $p < 0.1$.

The findings indicate that the positive impact of digital development on port security resilience has indeed persisted over the long term, with a slight diminishing trend. This sustained positive effect can be attributed to the gradual implementation and scaling up of technologies such as big data, artificial intelligence, and digital twins in port operations. As digitization progresses, the synergies of these technologies are maximized, with historical data playing a crucial role in preventing safety incidents and ensuring port security. This gradual buildup and optimization of digital constructs explain the long-term impact of digital development on port security resilience observed in this study.

## 5. Conclusions and Future Research

### 5.1. Conclusions

This study gathered data from provincial panels in China's coastal provinces and select provinces of the Yangtze River Economic Belt spanning from 2010 to 2019. Utilizing the entropy weight method, the researchers developed the port security resilience level evaluation index (RES) and the digital development level evaluation index (DIG) to investigate the influence of digital development on port security resilience. The significance of this study is in providing a data- and evidence-based scientific foundation for enhancing port security resilience, especially in the current era of rapid digital evolution. By examining the impact of digital development on port security resilience, this research not only provides policymakers with specific guidance, but also offers actionable strategies for port authorities and related businesses to effectively utilize digital technologies for enhancing port safety management and response capabilities. The empirical findings are as follows. First, digital development plays a crucial role in enhancing the resilience of port security, with a more pronounced impact on coastal provinces compared to inland provinces. This intricate process not only drives societal and economic growth but also fosters interconnectivity among various industries. By leveraging digital technologies like the Internet of Things, ports can establish seamless communication channels between different sectors, including port operations, logistics, and shipping. Furthermore, digital advancements enhance port safety through real-time ship-monitoring systems that minimize collisions and automated identification systems for container cranes that improve operational efficiency and reduce the risk of accidents. Additionally, technologies like 5G and cloud storage enable ports to store and analyze vast amounts of data, empowering decision-making processes during times of uncertainty or risk.

Second, focusing on personnel training, scientific and technological development, and social investment can enhance the resilience of port security. Digital development relies on talent support, government investment, and a focus on scientific and technological progress. The 20th Report of the Communist Party of China emphasizes the importance of accelerating the digital economy, prioritizing talent as a key resource, and supporting talent for modernization. The EU 2020 Action Plan for Digital Education aims to enhance education and training systems in EU member states for the digital era, with initiatives such as updating digital competence frameworks, establishing digital education centers, and creating a European Certificate of Digital Skills. The National Security Council's 2021 Final Report on Artificial Intelligence highlights the United States' lack of preparedness in training artificial intelligence talent and fostering digital literacy among its population, potentially impacting industrial competitiveness and national security [51]. It is clear that nations worldwide are placing importance on training digital talent and advancing science and technology. Social investment plays a key role in improving social development conditions, energizing social progress, and driving digital advancements.

Third, in the analysis of long-term effects, digital development has a lasting positive impact on port security resilience, although it shows a slightly diminishing trend over time. This suggests that the benefits of enhancing port security resilience through digital development will eventually plateau, emphasizing the importance for local governments to actively engage in digital initiatives to promptly capitalize on efficiency gains. Especially in the post-pandemic era, as the global economy rebounds and ports become busier worldwide, addressing port congestion through digital advancements to improve operational efficiency is crucial. This not only alleviates port congestion but also enhances safety for life and property.

In consideration of the findings above, the following recommendations are put forth:

Policymakers should prioritize increasing investments in digital infrastructure, including 5G networks, cloud computing services, and Internet of Things (IoT) technologies. It is crucial to reform the education system to enhance the development of digital skills and talent in artificial intelligence, ensuring a strong workforce for digital transformation. Additionally, promoting information sharing and collaboration among ports, logistics, and

shipping companies through a unified digital platform is essential for optimizing resource allocation and improving operational efficiency. Policymakers should also create policies to incentivize ports to establish emergency response mechanisms and disaster recovery plans, utilizing digital technologies for simulation drills to ensure swift and effective responses to emergencies. Regularly updating and refining policies and regulations related to digital development is vital to provide legal support for port safety resilience, while also promoting the adoption of digital certificates, blockchain, and other technologies to enhance the transparency and traceability of port operations.

For port authorities, it is advisable to actively implement digital tools such as intelligent monitoring systems and automated container management systems to reduce human error, thereby enhancing operational efficiency and safety. Collaboration with higher education institutions and research organizations to advance research and application of port-related digital technologies is crucial for driving the adoption of innovative solutions. Strengthening coordination with other inland and coastal ports to share safety information and address broad safety challenges and risk management is also advised. Establishing and refining intelligent early warning systems and utilizing big data analysis to predict potential risks and respond promptly to mitigate the impact of disasters is essential. Port authorities should proactively adjust to policy changes and conduct regular safety evaluations and compliance checks to ensure that technology applications and data processing align with the latest legal and regulatory requirements.

*5.2. Limitations and Future Research*

The establishment of definitive standards for defining port resilience remains a challenge within the academic community due to the complexity and dynamism of the concept. This paper proposes a definition of port safety resilience that focuses on water traffic safety within port jurisdiction. This approach allows for the measurement of resilience using objective quantitative data, although it does not encompass factors such as natural disasters, cyber attacks, and emergencies that can impact port resilience. Future research should explore these unaddressed areas. Additionally, in light of the global impact of the COVID-19 pandemic, the study period was set between 2010 and 2019 to mitigate the influence of extreme circumstances on research results. Investigating the resilience of port security in the context of the pandemic presents an important avenue for future research.

**Author Contributions:** Conceptualization, X.R. and J.S.; methodology, J.S.; software, J.S.; validation, J.S. and Z.F.; formal analysis, J.S.; investigation, J.S. and Z.F.; resources, J.S. and Z.F.; data curation, J.S. and Z.F.; writing—original draft preparation, J.S.; writing—review and editing, X.R. and J.S.; visualization, X.W.; supervision, X.R. and J.S.; project administration, K.A.; funding acquisition, X.R. All authors have read and agreed to the published version of the manuscript.

**Funding:** This work is financially supported by the China National Natural Science Foundation: 21BJY223; Chongqing Natural Science Foundation Project: CSTB2023NSCQ-MSX0046; Major Project of Chongqing Social Science Planning "Construction of New Land and Sea Passage in the West": 2023ZDLH06.

**Institutional Review Board Statement:** Not applicable.

**Informed Consent Statement:** Not applicable.

**Data Availability Statement:** Data will be made available on request.

**Conflicts of Interest:** The authors declare that they have no known competing financial interests or personal relationships that could have appeared to influence the work reported in this paper.

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
