# Peer review of "The Impact of Digital Development on Port Security Resilience—An Empirical Study from Chinese Provinces"

_sustainability, doi:10.3390/su16062385_

Round 1

Reviewer 1 Report

Comments and Suggestions for Authors

The paper introduces a comprehensive RES index system and a DIG evaluation system, incorporating three crucial perspectives for port safety resilience. The use of the entropy weight method adds rigor to the evaluation process. 

1. More details on the methodology, such as data sources and specific criteria for evaluation, would enhance the paper's transparency.

2. The decision to focus on 16 provinces in China's coastal and Yangtze River Economic Belt provides a valuable regional perspective. The use of provincial panel data from 2010 to 2019 allows for an in-depth analysis. However, the paper could benefit from a brief discussion on the selection criteria for these provinces.

3. How can this information be used to enhance port security measures? Are there specific recommendations for policymakers or port authorities?

4. To enhance the paper's impact, the authors could discuss the generalizability of their findings to other regions or countries, providing insights into the broader applicability of their research.

In summary, the paper addresses an important issue in post-pandemic port activities and digital development. To enhance the manuscript, the authors are encouraged to provide more details on the methodology, discuss the broader implications of their findings, and offer a more nuanced analysis of regional heterogeneity and long-term effects.

Comments on the Quality of English Language

 Moderate editing of English language required

Author Response

Upload attachments only.

Reviewer 2 Report

Comments and Suggestions for Authors

This study analyzes the relationship between digital development and port safety resilience and has a very high level of creativity. In particular, equations (1) to 5 are novel methodologies. In addition, the fact that a long-term effect test method was performed through extended analysis starting from line 382 enhances the reliability of the authors' research results.

Author Response

Upload attachments only.

Reviewer 3 Report

Comments and Suggestions for Authors

The authors have done a lot of work and conducted statistical analysis. But I am confused about the selection of this indicator, what is the significance of port digital technology for production, and how does your data reflect the value of your manuscript?

Comments on the Quality of English Language

need to be edited by native speaker if possible

Author Response

Upload attachments only.

Reviewer 4 Report

Comments and Suggestions for Authors

The topic of the article is very relevant. But there are some issues that the authors need to take into consideration.

1. At the end of the Introduction section, the authors annotate the article. In my opinion, this section should end with an exposition  of the arguments in favour of the relevance of the topic;

2. The analysis of literary sources should be strengthened, namely, to emphasise the unsolved problem. I.e. critical analysis should be strengthened.

3. The inscriptions in Figures 1 – 5 are very small and almost unreadable.

4. References to literary sources in the text of the article are very inconvenient. I recommend the authors to provide references to the relevant sources in the text of the article in numerical form in square brackets.

5. The conclusions are more like a Discussion. I recommend that the authors revise them. It may be worth adding the Discussion section to the article, and structuring the conclusions in accordance with the objectives.

Author Response

Upload attachments only.

Reviewer 5 Report

Comments and Suggestions for Authors

The article has great impact on ensuring traffic safety within port jurisdictions. Their work examines the influence of digital development on port security resilience. In which they evaluate provincial digital development from the perspective of digital infrastructure, subsequently observing rescue and recovery capabilities and operational efficiency, obtaining as results that digital development improves the resilience of port security, observing significant heterogeneity and long-term effects. . However, the article presents observations that must be attended to.

1.- It is necessary to include a paragraph of the contribution of this work. This should go in the introduction section.

2.- You must write a short paragraph about what section 2 “Literature Review” will discuss before starting section 2.1. “Resilience and Port Security”.

3..-A short paragraph must be written about what section 3 “Materials and Methods” will cover before starting section 3.1. “Study Area”.

4.- check the aforementioned in sections 4 and 5.

5.- Figures 1-5 do not have the image quality, it is necessary to improve them and the size of the letters.

6.- What specific improvements should the authors consider regarding the methodology? What further controls should be considered?.

7.- Are the conclusions consistent with the evidence and arguments presented and do they address the main question posed?

Comments on the Quality of English Language

The article has great impact on ensuring traffic safety within port jurisdictions. Their work examines the influence of digital development on port security resilience. In which they evaluate provincial digital development from the perspective of digital infrastructure, subsequently observing rescue and recovery capabilities and operational efficiency, obtaining as results that digital development improves the resilience of port security, observing significant heterogeneity and long-term effects. . However, the article presents observations that must be attended to.

1.- It is necessary to include a paragraph of the contribution of this work. This should go in the introduction section.

2.- You must write a short paragraph about what section 2 “Literature Review” will discuss before starting section 2.1. “Resilience and Port Security”.

3..-A short paragraph must be written about what section 3 “Materials and Methods” will cover before starting section 3.1. “Study Area”.

4.- check the aforementioned in sections 4 and 5.

5.- Figures 1-5 do not have the image quality, it is necessary to improve them and the size of the letters.

6.- What specific improvements should the authors consider regarding the methodology? What further controls should be considered?.

7.- Are the conclusions consistent with the evidence and arguments presented and do they address the main question posed?

Author Response

Upload attachments only.

Round 2

Reviewer 1 Report

Comments and Suggestions for Authors

I really appreciate the authors' revisions, which effectively address the issues that were raised.

Author Response

Thank you for your valuable comments. Your comments have greatly improved the quality of my manuscripts.

Reviewer 5 Report

Comments and Suggestions for Authors

All observations were attended, I have no more observations.

Comments on the Quality of English Language

All observations were attended, I have no more observations.

Author Response

Thank you for your valuable comments. Your comments have greatly improved the quality of my manuscripts. For the language problem, I have made a lot of changes.